Comparative analysis of the complete chloroplast genome sequences in psammophytic Haloxylon species (Amaranthaceae)

Dong Wenpan 1 2
Xu Chao 1 3
Li Delu 4
Jin Xiaobai 5
Li Ruili 5
Lu Qi Luqi@caf.ac.cn 6
Suo Zhili zlsuo@ibcas.ac.cn 1
1 State Key Laboratory of Systematic and Evolutionary Botany, Institute of Botany, Chinese Academy of Sciences , Beijing , China
2 Peking-Tsinghua Center for Life Sciences, Academy for Advanced Interdisciplinary Studies, Peking University , Beijing , China
3 University of Chinese Academy of Sciences , Beijing , China
4 Gansu Desert Control Research Institute , Gansu , China
5 Beijing Botanical Garden, Institute of Botany, Chinese Academy of Sciences , Beijing , China
6 Institute of Desertification Studies, Chinese Academy of Forestry , Beijing , China
Venancio Thiago
Electronic publication date: 2016 Nov 10
Publication date: 2016
Volume: 4
Electronic Location ID: e2699
Received 2016 Aug 2; Accepted 2016 Oct 18
Copyright: ©2016 Dong et al.
Copyright year: 2016
Copyright holder: Dong et al.
License: This is an open access article distributed under the terms of the Creative Commons Attribution License, which permits unrestricted use, distribution, reproduction and adaptation in any medium and for any purpose provided that it is properly attributed. For attribution, the original author(s), title, publication source (PeerJ) and either DOI or URL of the article must be cited.
License URL: https://creativecommons.org/licenses/by/4.0/

Keywords: Chloroplast genome, Psammophytes, Structure, Evolution, Amaranthaceae, Haloxylon

Funding: China National Science and Technology Plan Project 2012BAD16B0101 948 Program 2008-4-47 Identification and Screening of Haloxylon Germplasm Resources 80117B1001 State Forestry Administration “Public Welfare Research Foundation” 201004010 Chinese Ministry of Science and Technology 2006BAD26B0101 National Natural Science Foundation of China 30972412 Joint projects 70009C1020 70009C1036 National Forest Genetic Resources Platform 2005DKA21003 This study was financially supported by the China National Science and Technology Plan Project (2012BAD16B0101), the “948 Program” (No. 2008-4-47), the Identification and Screening of Haloxylon Germplasm Resources (80117B1001), the State Forestry Administration “Public Welfare Research Foundation” (No. 201004010), the Scientific and Technological Support Project from the Chinese Ministry of Science and Technology (2006BAD26B0101), the National Natural Science Foundation of China (No. 30972412), the joint projects No. 70009C1020 and 70009C1036, and the National Forest Genetic Resources Platform (2005DKA21003). The funders had no role in study design, data collection and analysis, decision to publish, or preparation of the manuscript.

==============================
The Haloxylon genus belongs to the Amaranthaceae (formerly Chenopodiaceae) family. The small trees or shrubs in this genus are referred to as the King of psammophytic plants, and perform important functions in environmental protection, including wind control and sand fixation in deserts. To better understand these beneficial plants, we sequenced the chloroplast (cp) genomes of Haloxylon ammodendron (HA) and Haloxylon persicum (HP) and conducted comparative genomic analyses on these and two other representative Amaranthaceae species. Similar to other higher plants, we found that the Haloxylon cp genome is a quadripartite, double-stranded, circular DNA molecule of 151,570 bp in HA and 151,586 bp in HP. It contains a pair of inverted repeats (24,171 bp in HA and 24,177 bp in HP) that separate the genome into a large single copy region of 84,214 bp in HA and 84,217 bp in HP, and a small single copy region of 19,014 bp in HA and 19,015 bp in HP. Each Haloxylon cp genome contains 112 genes, including 78 coding, 30 tRNA, and four ribosomal RNA genes. We detected 59 different simple sequence repeat loci, including 44 mono-nucleotide, three di-nucleotide, one tri-nucleotide, and 11 tetra-nucleotide repeats. Comparative analysis revealed only 67 mutations between the two species, including 44 substitutions, 23 insertions/deletions, and two micro-inversions. The two inversions, with lengths of 14 and 3 bp, occur in the petA-psbJ intergenic region and rpl16 intron, respectively, and are predicted to form hairpin structures with repeat sequences of 27 and 19 bp, respectively, at the two ends. The ratio of transitions to transversions was 0.76. These results are valuable for future studies on Haloxylon genetic diversity and will enhance our understanding of the phylogenetic evolution of Amaranthaceae.

Introduction

The eudicot clade comprises approximately 75% of all flowering land plant species, including major subclades: rosids, asterids, Saxifragales, Santalales, and Caryophyllales (APG III, 2009). Haloxylon species, which include psammophytic small trees or shrubs, are positioned phylogenetically in the Amaranthaceae Juss of the Caryophyllales Perleb among core eudicots (APG III, 2009; Pyankov et al., 2001; Akhani, Edwards & Roalson, 2007). The Haloxylon genus has about 11 species, with a distribution from the Mediterranean through Central Asia and into China (Zhu, Mosyakin & Clemants, 2004). Two Haloxylon species, which are known as the King of psammophytic plants, are found in the deserts of northwest China and, play important roles in environmental protection, including wind control and sand fixation (Zhu, Mosyakin & Clemants, 2004; Jia & Lu, 2004). These precious psammophytic woody plants can adapt to harsh environmental conditions, such as drought, desert, high temperature, and sand storms. However, populations of Haloxylon plants have been threatened in China in past decades as a result of decreased underground water, overgrazing, and over exploitation of agriculture.

Because of the environmental significance of these plants and their declining numbers, genetic research on Haloxylon germplasm resources has garnered significant interest (Song & Jia, 2000; Sheng et al., 2004; Sheng et al., 2005; Zhang et al., 2006a; Zhang et al., 2006b). However, Haloxylon plants possess only fine green assimilating shoots, without leaves, making the evaluation of their phenotypic diversity difficult. Further, the detection of genetic diversity within Haloxylon germplasm resources has been slowed by a lack of morphological markers (Sheng et al., 2004; Sheng et al., 2005; Zhang et al., 2006a; Zhang et al., 2006b; Wang et al., 2009; Suo et al., 2012a). A recent study by Long et al. (2014) used RNA-seq data to elucidate the Haloxylon transcriptome, providing a valuable sequence resource for further genetic and genomic studies; however, genetic information for members of the Haloxylon genus, and how they might differ from one another, is limited.

Each leaf cell of plants contains 1,000–10,000 chloroplasts (cp), which are key organelles for photosynthesis and other biochemical pathways such as the biosynthesis of starch, fatty acids, pigments, and amino acids (Dong et al., 2013b; Raman & Park, 2016). Since the first cp genome of Nicotiana tabacum was sequenced in 1986, around 800 complete cp genome sequences have been made available in the National Center for Biotechnology Information organelle genome database. These data are valuable sources of genetic markers for phylogenetic analyses, genetic diversity evaluation, and plant molecular identification (Dong et al., 2012; Dong et al., 2013a; Dong et al., 2013b; Dong et al., 2014; Ni et al., 2016; Suo et al., 2012b).

There are two published complete cp genome sequences (Spinacia oleracea and Beta vulgaris subsp. vulgaris) from members of the Amaranthaceae family (Li et al., 2014; Schmitz-Linneweber et al., 2001). However, the determination of the cp genome from Haloxylon plants is of further significance for potentially enhancing our understanding of their adaptability to severe desert environmental conditions, and their genomic evolution within the Amaranthaceae. Here, we report the complete cp genomes from two Haloxylon species, H. ammodendron and H. persicum, including patterns of nucleotide substitutions, microstructural mutation, and simple sequence repeats (SSRs). We further performed genomic comparative analyses on these and two other representative Amaranthaceae species, to better understand the evolutionary relationships within this family.

Materials & Methods

Sampling and DNA extraction

Fresh young shoots of H. ammodendron (HA) and H. persicum (HP) were collected in May 2011 from Minqin Eremophytes Botanical Garden (N38°34′, E102°59′, Altitude 1,378 m), Gansu Province, China (under the leadership of Gansu Desert Control Research Institute, Lanzhou, Gansu, China). These HA and HP plants were originally introduced from the Turpan Desert Botanical Garden of Chinese Academy of Sciences, Xinjiang Uygur Autonomous Region. The shoots from each accession were immediately dried using silica gel for future DNA extraction. Total genomic DNA was extracted from each using the Plant Genomic DNA Kit (DP305) from Tiangen Biotech (Beijing) Co., Ltd., China. The approval numbers are 2012BAD16B0101 and 80117B1001 for field permit of the research.

Chloroplast genome sequencing

The HA and HP cp genomes were sequenced using the short-range PCR method reported by Dong et al. (2012), Dong et al. (2013a) and Dong et al. (2013b). The PCR protocol was as follows: preheating at 94 °C for 4 min, 34 cycles at 94 °C for 45 s, annealing at 55 °C for 40 s, and elongation at 72 °C for 1.5 min, followed by a final extension at 72 °C for 10 min. PCR amplification was performed in an Applied Biosystems VeritiTM 96-Well Thermal Cycler (Model#: 9902, made in Singapore). The amplicons were sent to Shanghai Majorbio Bio-Pharm Technology Co., Ltd. (Beijing) for Sanger sequencing in both the forward and reverse directions using a 3730xl DNA analyzer (Applied Biosystems, Foster City, CA, USA). DNA regions containing poly structures or that were difficult to amplify were further sequenced using newly designed primer pairs for confirming reliable and high quality sequencing results.

Chloroplast genome assembling and annotation

The cp DNA sequences were manually confirmed and assembled using Sequencher (v4.6) software, and cp genome annotation was performed using the Dual Organellar Genome Annotator (DOGMA) (Wyman, Jansen & Boore, 2004). BLASTX and BLASTN searches were utilized to accurately annotate the protein-encoding genes and to identify the locations of the transfer RNAs (tRNAs) and ribosomal RNAs (rRNAs). Gene annotation information from other closely related plant species was also used for confirmation when the boundaries of the introns or exons could not be precisely determined because of the limited power of BLAST in cp genome annotation (e.g., for some short exons of 6–9 nt in length, such as in the case of rps16, petB, and petD). Promoter, intron, and exon boundaries, as well as the location of stop codons for all protein-encoding genes, have been identified accurately. The cp genome map was drawn using Genome Vx software (Conant & Wolfe, 2008) (http://wolfe.ucd.ie/GenomeVx/), and the cp genome sequences have been deposited to GenBank with the following accession numbers: KF534478 for HA and KF534479 for HP (https://www.ncbi.nlm.nih.gov/nuccore/?term=Haloxylon+chloroplast+genome).

Repeat structure analysis

Gramene Simple Sequence Repeat Identification Tool software (http://www.gramene.org/db/markers/ssrtool) (Benson, 1999) was utilized to search for simple sequence repeat loci in the cp genome sequences, with the threshold value of repeat number as ≥10 for mono-nucleotide repeats, ≥5 for di-nucleotide repeats, ≥4 for tri-nucleotide repeats, and ≥3 for tetra-nucleotide, penta-nucleotide, or hexa-nucleotide repeats.

Gene content analysis and comparative genomics

The mVISTA program was employed in Shuffle-LAGAN mode (Frazer et al., 2004) to compare the complete HA and HP cp genomes. These were aligned using MUSCLE software (Thompson et al., 1997) and were manually adjusted using Se-Al 2.0 (Rambaut, 1996). Variable sites in the cp genome were calculated using DnaSP (DNA Sequences Polymorphism version 5.10.01) software (Librado & Rozas, 2009), and the genetic distance (p-distance) was computed using MEGA 6.0 software (Tamura et al., 2013). Based on the aligned sequence matrix, the micro-structure events were checked manually and were further divided into three categories: (i) microsatellite-related insertions/deletions (indels), (ii) non-microsatellite-related indels, (iii) and inverted sequences. Using the HA cp genome sequence as the standard reference, the size, location, and evolutionary direction of the microstructure events were counted. The proposed secondary structures of the inverted regions in the cp genomes of HA and HP were analyzed using mfold software (Zuker, 2003). The complete cp genome sequences of S. oleracea (GenBank accession number AJ400848.1, Spinacia L.) (Schmitz-Linneweber et al., 2001) and B. vulgaris subsp. vulgaris (GenBank accession number KJ081864.1, Beta vulgaris subsp. vulgaris) (Li et al., 2014), two closely related species in the Amaranthaceae family, were downloaded from GenBank databases (www.ncbi.nlm.nih.gov). These were used for comparison with the complete cp genomes of HA and HP.

Results & Discussion

Genome features

Similar to the typical cp genome structure in other higher plants, the Haloxylon cp genome is a double-stranded, circular DNA molecule of 151,570 bp in length in HA and 151,586 bp in length in HP. It also includes a large single copy region (LSC) of 84,214 bp in HA and 84,217 bp in HP and a small single copy region (SSC) of 19,014 bp in HA and 19,015 bp in HP; these are separated by a pair of inverted repeats (IR) (24,171 bp in HA and 24,177 bp in HP) (Fig. 1). The GC content in this IR region is 43.0% in HA and 42.7% in HP, and the GC content in the LSC and SSC regions is 34.4% (LSC) and 29.7% (SSC) in HA and 34.5% (LSC) and 29.7% (SSC) in HP (Table 1).

Figure 1 Representative map of the two Haloxylon chloroplast genomes.

Genome annotation was performed using DOGMA. Genes drawn outside of the circle are transcribed clockwise, whereas those represented inside the circle are transcribed counterclockwise. Small single copy (SSC), large single copy (LSC), and inverted repeat (IRa, IRb) regions are indicated.

Table 1 Summary of complete chloroplast genome features in Haloxylon.

	H. ammodendron	H. persicum	Spinacia oleracea	Beta vulgaris	
Total cpDNA size	151,570	151,586	150,725	149,635	
Length of LSC region	84,214	84,217	82,719	83,057	
Length of IR region	24,171	24,177	25,073	24,439	
Length of SSC region	19,014	19,015	17,860	17,701	
Total GC content (%)	36.6	36.6	36.9	36.4	
LSC	34.4	34.5	34.8	34.1	
IR	43.0	43.0	42.7	42.2	
SSC	29.7	29.7	29.8	29.2	
Total number of genes	112	112	112	113	
Protein encoding	78	78	78	79	
tRNA	30	30	30	30	
rRNA	4	4	4	4	
Pseudogenes	2	2	2	1	

Among the four Amaranthaceae species included in our analyses, which represent three genera, the longest cp genomes (151,570 bp for HA and 151,586 bp for HP) are 1,935 bp to 1,951 bp larger than the shortest one (149,635 bp for B. vulgaris subsp. vulgaris) (Li et al., 2014). The size of the S. oleracea cp genome (150,725 bp) (Schmitz-Linneweber et al., 2001) is intermediate (Table 1). Notably, the cp genomes of HP and HA are quite similar in size; the HP cp is only 16 bp longer than that of HA, with minor differences between them.

There are a total of 112 genes in the Haloxylon cp genome, including 78 coding genes, 18 of which are duplicated genes in the IR region, 30 tRNA genes, and four ribosomal RNA genes (16S, 23S, 5S, 4.5S) (Fig. 1 and Table S1). Based on their predicted functions, these genes can be divided into three categories, (1) genes related to transcription and translation; (2) genes related to photosynthesis; (3) genes related to the biosynthesis of amino acids, fatty acids, etc., and some functionally unknown genes (Table S1). The S. oleracea cp also contains the same 78 protein-coding genes, whereas the cp in B. vulgaris has 79. This species contains an additional gene (rpl23), which is a pseudogene in the other species (Fig. 1 and Table S1). There are 17 genes harboring introns in the cp genomes of the four Amaranthaceae species analyzed (one class I intron, trnL-UUA, and 16 class II introns), and two of these genes, ycf3 and clpP, contain two introns each (Table 2).

Table 2 Genes with introns in Haloxylon ammodendron and H. persicum and length of exons and introns.

	Exon I (bp)	Intron I	Exon II	Intron II	Exon III	
atpF	145(145)	785(784)	410(410)			
clpP	71(71)	951(951)	292(292)	601(601)	228(228)	
ndhA	553(553)	1090(1090)	533(533)			
ndhB	777(777)	675(675)	756(756)			
petB	6(6)	801(801)	642(642)			
petD	8(8)	722(722)	475(475)			
rpl16	399(399)	913(913)	9(9)			
rpl2	393(393)	668(668)	435(435)			
rpoC1	432(432)	780(780)	1602(1602)			
rps12	114(114)	–	231(231)	–	27(27)	
rps16	40(40)	881(881)	197(197)			
trnA-UGC	38(38)	831(831)	42(42)			
trnG-GCC	23(23)	722(722)	58(58)			
trnI-GAU	42(42)	942(941)	35(35)			
trnK-UUU	35(35)	2909(2909)	37(37)			
trnL-UAA	35(35)	557(557)	50(50)			
trnV-UAC	39(39)	602(602)	35(35)			
ycf3	126(126)	772(772)	229(229)	812(812)	152(152)	
Notes.

rps12 is trans-spliced with the 5′end located in the LSC region and the duplicated 3′end in the IR regions.

Several angiosperm lineages have lost introns from the rpl2 gene independently (Downie et al., 1991), which could also be regarded as a characteristic feature of the core members of the Caryophyllales (Logacheva et al., 2008). In each of the four Amaranthaceae cp genomes in our analysis, the rpl2 gene has lost its intron. Some authors have proposed that intron loss is not always a dependable marker of phylogenetic relationships (Millen et al., 2001; Dong et al., 2013b; Raman & Park, 2016), and further study, including the sampling of more taxa, is needed to clarify this issue.

Expansion and contraction of the border regions in Haloxylon cp genomes

To analyze these Amaranthaceae species at the genome-level, the sequences of all the four cp genomes were plotted using the VISTA program (Frazer et al., 2004), using the annotation of HA as a reference (Fig. 2). Similar to other angiosperms, we observed that the IR region is more conserved in these species than the LSC and SSC regions.

Figure 2 Identity plot comparing the chloroplast genomes of four Amaranthaceae species using Haloxylon ammodendron as a reference sequence.

The vertical scale indicates the percent identity, ranging from 50% to 100%. The horizontal axis indicates the coordinates within the chloroplast genome. Genomic regions are color coded as protein-coding, rRNA, tRNA, intron, and conserved non-coding sequences (CNS). Abbreviations HP, H. persicum; SO, Spinacia oleracea; BV, Beta vulgaris subsp. vulgaris.

The expansion and contraction of the border regions between the two IR regions and the single copy region have contributed to genome size variations among plant lineages (Dong et al., 2013b; Goremykin et al., 2003; Ni et al., 2016). Therefore, we next compared the exact IR border positions and their adjacent genes among the four Amaranthaceae cp genomes (Fig. 3). From these data, we see that the IRa/LSC border is generally located upstream of the trnH-GUG gene. The distance between the IRa/LSC border and the trnH-GUG gene is 1 bp in the Haloxylon cp genomes and 2 bp in Beta genus, with no separation in Spinacia (Fig. 3). The IR region is expanded by 763 bp and enters the 5′ end of the ycf1 gene in Haloxylon species, whereas it is expanded by 1,427 bp and 1,492 bp, respectively, in Spinacia and Beta. Except for the expansion of the ycf1 gene, the IR region extends to the rps19 gene in all of four Amaranthaceae cp genomes. The rps19 pseudogene was not observed in this study. Although there are expansions or contractions of IR regions observed among the investigated species of the Amaranthaceae, they contribute little to the overall size differences in the cp genomes. The exon at the 5′ end of the rps12 gene is located in the LSC region, and the intron and 3′-end exon of the gene are situated in the IR region in all four Amaranthaceae species.

Figure 3 Comparison of the junction positions between the single copy and IR regions among four Amaranthaceae genomes.

Indels and SNPs

Indel and single nucleotide polymorphism (SNP) sites are important molecular features valuable for development of DNA markers that are useful for plant identification and genetic analysis of population structure (Dong et al., 2012; Dong et al., 2013a; Dong et al., 2013b; Dong et al., 2014; Suo et al., 2012b; Suo et al., 2015; Suo et al., 2016). We detected 23 indels in the cp genome sequence alignment of HA and HP, including 16 indels caused by microsatellite repeat variations and seven non-microsatellite-related indels (Table 3). Most of the indel events occurred in non-coding regions (21/23). A large portion of the indels related to microsatellite repeat variations are characterized by a single base mutation; six insertions of this type were observed in the HA cp genome. The non-microsatellite-related indels were found to contain mostly five to six variable base sites, and two insertions of this type were detected in the HA cp genome.

Table 3 Indel mutation events in the chloroplast genomes of Haloxylon ammodendron and H. persicum.

Region	Location	Types	HA	HP	Length (bp)	Directiona	
accD-psaI	Intergenic	Homopolymeric indel	A A	–	2	Insertion	
atpA-atpF	Intergenic	Homopolymeric indel	T	–	1	Insertion	
atpF	Intron	Homopolymeric indel	–	T	1	Deletion	
ndhI-ndhA	Intergenic	Homopolymeric indel	–	A	1	Deletion	
ndhJ-ndhK	Intergenic	Homopolymeric indel	–	T	1	Deletion	
psbI-trnS	Intergenic	Homopolymeric indel	–	T	1	Deletion	
psbI-trnS	Intergenic	Homopolymeric indel	–	A	1	Deletion	
rbcL-accD	Intergenic	Homopolymeric indel	–	A	1	Deletion	
rps18-rpl20	Intergenic	Homopolymeric indel	T	–	1	Insertion	
trnE-trnT	Intergenic	Homopolymeric indel	–	A	1	Deletion	
trnK-rps16	Intergenic	Homopolymeric indel	A	–	1	Insertion	
trnK-rps16	Intergenic	Homopolymeric indel	A	–	1	Insertion	
trnL	Intron	Homopolymeric indel	–	A	1	Deletion	
trnL	Intron	Homopolymeric indel	A	–	1	Insertion	
trnL	Intron	Homopolymeric indel	–	T	1	Deletion	
trnR-aptA	Intergenic	Homopolymeric indel	–	T	1	Deletion	
atpH-atpI	Intergenic	Indel	T T T A T T	–	5	Insertion	
clpP-psbB	Intergenic	Indel	–	G T C T T	5	Deletion	
petL-petG	Intergenic	Indel	–	G	1	Deletion	
rpoB-trnC	Intergenic	Indel	–	T G T A T T	5	Deletion	
rpoB-trnC	Intergenic	Indel	T A C A A	–	5	Insertion	
rrn23	Coding	Indel	–	A A T T A A	6	Deletion	
rrn23	Coding	Indel	–	T T A A T T	6	Deletion	
Notes.

a The chloroplast genome of H. ammodendron was used as a standard.

HA H. ammodendron

HP H. persicum

Table 4 The nucleotide substitution patterns present in the two Haloxylon chloroplast genomes.

Region	Location	H. ammodendron	H. persicum	
atpA	Coding	G	A	
atpI	Coding	T	C	
matK	Coding	C	A	
ndhF	Coding	C	T	
ndhI	Coding	G	T	
psbC	Coding	A	C	
rpoB	Coding	C	T	
rpoC2	Coding	C	A	
rpoC2	Coding	C	G	
rpoC2	Coding	G	T	
rps15	Coding	A	G	
rps3	Coding	T	G	
ycf1	Coding	A	G	
ycf1	Coding	G	C	
ycf1	Coding	G	T	
atpB-rbcL	Intergenic	A	C	
atpF-atpH	Intergenic	G	C	
atpH-atpI	Intergenic	G	A	
ndhF-rpl32	Intergenic	G	T	
psaJ-rpl33	Intergenic	C	T	
psaJ-rpl33	Intergenic	T	A	
psbE-petL	Intergenic	C	A	
psbM-trnD	Intergenic	A	G	
rpl14-rpl16	Intergenic	T	G	
rpl20-rps12	Intergenic	G	T	
rpl33-rps18	Intergenic	T	C	
rpoA-rps11	Intergenic	A	G	
rpoA-rps11	Intergenic	T	C	
rpoB-trnC	Intergenic	G	T	
rpoB-trnC	Intergenic	T	G	
rps18-rpl20	Intergenic	T	G	
rps8-rpl14	Intergenic	G	A	
trnG-trnR	Intergenic	A	C	
trnH-psbA	Intergenic	T	G	
trnK-matK	Intergenic	A	C	
trnK-rps16	Intergenic	A	C	
trnP-psaJ	Intergenic	C	T	
trnP-psaJ	Intergenic	C	T	
clpP	Intron	T	G	
ndhA	Intron	T	C	
rpl16	Intron	T	C	
rps16	Intron	T	G	
trnV	Intron	T	C	
ycf3	Intron	T	C	

Forty-four SNPs were detected in the HA and HP cp genomes (Table 4), which is considerably less than what was found between the cp genomes of other closely related plant species, including Oryza sativa and Oryza nivara (159 SNPs, Masood et al., 2004), Machilus yunnanensis and Machilus balansae (231 SNPs, Song et al., 2015), Citrus sinensis and Citrus aurantiifolia (330 SNPs, Su et al., 2014), Panax ginseng and Palax notoginseng (464 SNPs, Dong et al., 2014), and Solanum tuberosum and Solanum bulbocastanum (591 SNPs, Chung et al., 2006). Of note, the indel and SNP mutation events in the Haloxylon cp genomes were not randomly distributed, but rather, clustered as “hotspots” (Shaw et al., 2007; Worberg et al., 2007). It is likely that such mutational dynamics created the highly variable regions in the genome (Suo et al., 2012b; Song et al., 2015).

Patterns of nucleotide substitutions

Overall, the differences between the HA and HP cp genomes are minor, with a genetic distance of 0.00029 between them (Table 4). In total, 44 variable nucleotide sites were detected, 23 of which were found in intergenic regions, six in introns, and 15 in protein-encoding regions.

We also found that the probability of occurrence for the various nucleotide substitutions is different, depending on the mutation, as shown in Fig. 4. The most frequently occurring mutations are from A to C and from T to G (12 times each); mutations from A to T and from T to A exhibited the lowest frequency (only one occurrence of each). The ratio of transitions (Ts) and transversions (Tv) was 0.76 in the cp genome of Haloxylon species.

Figure 4 The nucleotide substitution patterns in the two Haloxylon chloroplast genomes.

The patterns were divided into six types, as indicated by the six non-strand-specific base-substitution types (i.e., numbers of G to A and C to T sites for each respective set of associated mutation types). The H. ammodendron chloroplast genome was used as a standard.

In the gene-encoding regions of the HA and HP cp genomes, a total of 15 variable base sites were detected in 11 protein-encoding genes. Specifically, we found one mutation in each of the following genes: atpA, atpI, matK, ndhF, ndhI, psbC, rpoB, rps15, and rps3. Two genes, rpoC2 and ycf1, each contained three mutation sites (Table 5). These mutations included six Ts and nine Tv. Ten nonsynonymous substitutions occurred simultaneously in seven genes (Table 5).

Table 5 Comparison of the mutational changes, number of transitions (Ts) and transversions (Tv), and synonymous (S) and nonsynonymous (N) substitutions per protein-coding chloroplast gene in Haloxylon ammodendron and H. persicum.

Gene	Ts	Tv	S	N	
atpA	1	0	1	0	
atpI	1	0	1	0	
matK	0	1	0	1	
ndhF	1	0	0	1	
ndhI	0	1	0	1	
psbC	0	1	1	0	
rpoB	1	0	1	0	
rpoC2	0	3	0	3	
rps15	1	0	0	1	
rps3	0	1	0	1	
ycf1	1	2	1	2	
Total	6	9	5	10	

Table 6 Location of repeats in the Haloxylon ammodendron chloroplast genome.

No.	Location	Motif	No. of repeats	SSR start	SSR end	
1	trnK-matK	A	11	1,658	1,668	
2	trnK-rps16	A	12	4,210	4,221	
3	rps16-trnQ	A	10	6,461	6,470	
4	trnQ-psbK	A	10	6,957	6,966	
5	psbK-psbI	A	10	7,578	7,587	
6	psbI-trnS	A	12	7,854	7,865	
7	atpF intron	A	10	12,476	12,485	
8	rpoC1 intron	A	10	22,386	22,395	
9	trnE-trnT	A	10	31,169	31,178	
10	trnL-intron	A	12	47,464	47,475	
11	trnF-ndhJ	A	10	48,982	48,991	
12	rbcL-accD	A	12	57,323	57,334	
13	accD-psaI	A	10	59,584	59,593	
14	psbF	A	10	64,309	64,318	
15	clpP intron	A	10	71,717	71,726	
16	petB intron	A	18	75,505	75,522	
17	ndhI-ndhA	A	10	118,705	118,714	
18	psaA	C	10	40,165	40,174	
19	trnK-rps16	T	10	4,464	4,473	
20	psbI-trnS	T	10	7,745	7,754	
21	trnR-atpA	T	11	9,948	9,958	
22	atpA-atpF	T	10	11,532	11,541	
23	atpF intron	T	11	12,457	12,467	
24	rps2-rpoC2	T	11	15,957	15,967	
25	rps2-rpoC2	T	11	18,156	18,166	
26	rpoB	T	10	25,865	25,874	
27	trnD-trnY	T	10	30,323	30,332	
28	trnL-trnF	T	10	48,029	48,038	
29	ndhJ-ndhK	T	10	49,646	49,655	
30	trnV intron	T	15	52,214	52,228	
31	trnM-atpE	T	10	52,658	52,667	
32	rbcL-accD	T	14	57,377	57,390	
33	petL-petG	T	10	66,141	66,150	
34	psaJ-rpl33	T	12	67,499	67,510	
35	rps18-rpl20	T	10	68,447	68,456	
36	rpoA	T	10	78,219	78,228	
37	rps11-rpl36	T	12	79,577	79,588	
38	rpl32-trnL	T	11	11,2371	11,2381	
39	ndhA intron	T	12	119,581	119,592	
40	ndhA intron	T	10	119,793	119,802	
41	ycf1	T	12	125,285	125,296	
42	ycf1	T	10	125,890	125,899	
43	ycf1	T	14	126,895	126,908	
44	ycf1	T	10	127,195	127,204	
45	rps16-trnQ	A T	5	6,277	6,286	
46	trnS-trnG	A T	5	8,177	8,186	
47	trnS-trnG	A T	5	8,300	8,309	
48	trnN-ndhF	T A A	4	109,380	109,391	
49	psbA-trnK	T T G T	3	1,522	1,533	
50	matK-trnK	T T C T	3	3,873	3,884	
51	atpI-rps2	A T T A	3	15,121	15,132	
52	trnE-trnY	A T T A	3	31,084	31,095	
53	accD-psaI	T A A T	4	59,721	59,736	
54	rps18-rpl20	T T T A	3	68,474	68,485	
55	clpP intron	T T T C	3	71,598	71,609	
56	rrn23	A G G T	3	104,481	104,492	
57	trnL-ccsA	A A C C	3	113,312	113,323	
58	ycf1	T A A T	3	124,297	124,308	
59	rrn23	C TA C	3	131,310	131,321	

Repeat structure feature

Simple sequence repeats (SSRs) are also called microsatellites. Within the cp genomes of HA and HP, 59 different SSR loci were detected. Of these, 44 loci are mono-nucleotide repeats, three are di-nucleotide repeats, one is a tri-nucleotide repeat, and 11 are tetra-nucleotide repeats; penta-nucleotide repeats or those containing a higher number of nucleotide repeats were not detected. Among the SSR loci detected, the most frequently observed repeats were A/ T and A T/ T A, accounting for 77.97% of the total number of SSR loci (Table 6). By comparison, in the cp genomes of M. yunnanensis and M. balansae, 36 SSR loci were identified (Song et al., 2015).

Inversions

Inversions are important events in the evolution of plant cp genomes. Smaller inversions are less frequent in these genomes, and they are generally associated with hairpins (Fig. 5). Most inversions are found in spacers and introns, and in most cases, the presence/absence of inversions is highly homoplastic during cp genome evolution (Kim & Lee, 2005; Catalano, Saidman & Vilardi, 2009), even at the population level (Quandt & Stech, 2004). A sequence alignment of the Haloxylon cp genomes revealed that an inversion event of 14 bp and one of 3 bp occur in the petA-psbJ intergenic region and in the rpl16 intron, respectively. The two inverted sequences are predicted to form secondary hairpin structures, with repeat sequences of 27 bp and 19 bp at the two ends, respectively (Fig. 5).

Figure 5 The hairpin loops predicted to be formed by inversions in the Haloxylon chloroplast genomes.

Pseudogenes

Pseudogenes have been defined as nonfunctional regions of genomic DNA that originally derived from functional genes (Balakirev & Ayala, 2003). These are evolutionary relics of functional components in the genome that provide important information regarding the history of the gene and genome evolution (Balakirev & Ayala, 2003; Zou et al., 2009; Choi & Park, 2015). The rpl22 and rps18 genes are putative pseudogenes in the Paeoniaceae (Dong et al., 2013b), whereas the atpB gene is a pseudogene in Aster spathulifolius. Conversely, the rpl22, rps18, and atpB genes are predicted to be normal and functional in the Haloxylon species, whereas rpl23 is a present as a pseudogene in the Haloxylon cp genomes (Fig. 1 and Table S1).

Conclusions

Two Haloxylon cp genomes were sequenced and characterized for the first time, and we found that they share the same overall organization and gene content found in most angiosperm cp genomes, including that of the closely related Spinacia and Beta species. The location and distribution of repeat sequences and differing nucleotide mutation sites between the two cp genomes were identified. The LSC/IRB/SSC/IRA boundary regions of the Amaranthaceae cp genomes were compared, and lightly intense variations were identified within the genus Haloxylon. The complete Haloxylon cp genome sequences reported here enhance the genomic information available for the Amaranthaceae family and further contribute to the study of germplasm diversity. These data represent a valuable source of markers for future research on Haloxylon population genetics.

Supplemental Information

Table S1 Genes found in the Haloxylon chloroplast genomes

Click here for additional data file.

The authors thank Prof. Borong Pan for advice and helpful discussion.

Additional Information and Declarations

Competing Interests

Author Contributions

Field Study Permissions

DNA Deposition

The authors declare there are no competing interests.

Wenpan Dong conceived and designed the experiments, performed the experiments, analyzed the data, wrote the paper, prepared figures and/or tables, reviewed drafts of the paper.

Chao Xu analyzed the data, wrote the paper, prepared figures and/or tables, reviewed drafts of the paper.

Delu Li contributed reagents/materials/analysis tools, reviewed drafts of the paper.

Xiaobai Jin wrote the paper, reviewed drafts of the paper.

Ruili Li prepared figures and/or tables, reviewed drafts of the paper.

Qi Lu conceived and designed the experiments, contributed reagents/materials/analysis tools, wrote the paper, reviewed drafts of the paper.

Zhili Suo conceived and designed the experiments, performed the experiments, analyzed the data, contributed reagents/materials/analysis tools, wrote the paper, reviewed drafts of the paper.

The following information was supplied relating to field study approvals (i.e., approving body and any reference numbers):

Minqin Eremophytes Botanical Garden, Gansu Province, China (under the leadership of Gansu Desert Control Research Institute, Gansu, China).

The approval numbers are 2012BAD16B0101 and 80117B1001.

The following information was supplied regarding the deposition of DNA sequences:

The chloroplast genome sequences have been deposited to GenBank with the following accession numbers: KF534478 for Haloxylon ammodendron and KF534479 for Haloxylon persicum.

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
