# Peer review of "Comparative analysis of the complete chloroplast genome sequences in psammophytic Haloxylon species (Amaranthaceae)"

_PeerJ, doi:10.7717/peerj.2699_

## Round 0.1 · original submission · Major Revisions

Please make sure to carefully check the points raised by Reviewers 2 (who has also supplied an annotated manuscript) and 3.

Reviewer 1 ·

Basic reporting

No Comments

Experimental design

No Comments

Validity of the findings

No Comments

Additional comments

The author of this manuscript has sequenced and assembled the complete chloroplast genomes of two Haloxylon species, which are well known as desert plants and play a very important role in environment protection in the deserts of China. They have compared the genomic structure and characterized the nucleotide substitutions between two Haloxylon species and two other Amaranthaceae species. The experimental design was validity and the findings are valuable for the future studies on their genetic diversity and evolutionary history.

·

Basic reporting

The authors have characterized the chloroplast genomes from two Haloxylon species. They reported the differences between Haloxylons and B. vulgaris and S. oleracea in terms of indels, SNPs, presence or absence of SSRs and the conservation of the whole chloroplasts. The background of the manuscript was well referenced and showed the context of the work. The figures were also relevant, although the need of a reordering. The raw data were supplied by the authors. The authors followed the structure of the journal PeerJ. However, the manuscript is lacking in some aspects and it requires revisions before it is considered for publication.
-The whole manuscript needs language improvement. The language feedback was added to the manuscript PDF.
-The results/discussion section needs an improvement to show the confrontation of results obtained with the data already published. Some sentences of the results also need to be rewritten to be clearer. I would suggest a completely revision of the topics' order in the result/discussion section. As example, the last topic "Expansion and contraction of the border regions in Haloxylon cp genomes" should be the second topic of the results.

Experimental design

The research question was clear, relevant and performed accordingly. The applied methods were well described and detailed. The article is in the scope of the journal PeerJ.

Validity of the findings

The data was robust and correctly deposited at NCBI. The conclusion presented the main findings of the manuscript, accordingly.

Reviewer 3 ·

Basic reporting

This manuscript clearly represents a prodigious effort to characterize and make comparative analysis of chloroplast genomes between the sequenced Haloxylon species and other ones in Amaranthaceae family. The work is interesting for the research and the environmental area and the results are valuable for studies on genetic diversity and for phylogenetic evolution analysis.

The English language and style are fine with minor spell check required, I would recommend to read the text carefully again.
For instance, wrong words like "inversionts" in line 43, or misleading sentences like: "...gene has lost its only intron...", in line 170.

The provided figures, tables and supplemental material are very informative, in very good resolution and are described appropriately, which improves the way to understand the work described in the manuscript.

Experimental design

The study and characterization of chloroplast genomes is important for the study of evolution and genetic diversity of plants, specially when they play an important role in environmental protection.
Since there were already sequenced chloroplast genomes for some species in the Amaranthaceae Family, the sequenced chloroplast genomes of the two Haloxylon species did not bring to much differences for the existing genomes, besides that, the genomes were fully sequenced and well characterized with different approaches, with particular molecular differences.

Specific comments and questions:

1 - About the sequencing, is there information of the genome sequencing coverage ? It is an important information to know about the power of the method and the quality of the assembled genome.

2 - For the genome annotation is very important to describe the parameters and threshold used in the genome/gene annotation, even if it was the default, such as:
- What is the DOGMA software last update ?
- What is the number of chloroplast genomes in the DOGMA database ?
- What were the e-value and % identity of protein and coding genes threshold ?

3 - In the analysis of micro-structure events,
- was the aligned sequences matrix obtained from MEGA software ?
- was used the default parameters ? if not, what were the parameters ?

Validity of the findings

1 - The characterization and description of findings in the genomes were well covered and compared with other species. The work provided relevant information about specific patterns, related to LSC/IRB/SSC/IRA genomic regions, indicating that they are probably related to the evolution of the species.

In relation to the Repeat structure analysis, I missed the use of more than one SSR tool, or the use of most common cited tools in the literature, such as MISA, SSRFinder or Sputnik, which could work as a confirmation of the predictors, since there are no experimental validation of the activity of predicted SSRs.

2 - The authors attempt for the lack of molecular markers resources for Haloxylon species, but there are an article of H. ammodendron species of RNA-seq for molecular marker studies, with strong overlap with this work, but was not cited:
"De novo assembly of the desert tree Haloxylon ammodendron (C. A. Mey.) based on RNA-Seq data provides insight into drought response, gene discovery and marker identification. Yan Long, Jingwen Zhang, Xinjie Tian, Shanshan Wu, Qiong Zhang, Jianping Zhang, Zhanhai DangEmail author and Xin Wu Pei. BMC Genomics, 2014. DOI: 10.1186/1471-2164-15-1111"

I would strongly recommend the authors to review this work and look carefully for the markers in genic regions in the above article, maybe the predictions can be validated by these results.

---

## Round 0.2 · Minor Revisions

Please just make the few additional corrections pointed by Reviewer 2.

·

Basic reporting

The authors have presented a better version of the manuscript with a language improvement. The figures and results section were also reordered. However, two changes are still needed:
-The results section "Patterns of nucleotide substitutions" should be placed after the "Indels and SNPs" section. This change will increase the text flow of the manuscript. The figures and tables number should be modified accordingly.
-The sentence in the line 199 needs revision - many "for" were used.

Experimental design

No comments.

Validity of the findings

No comments.

Reviewer 3 ·

Basic reporting

No Coments

Experimental design

No Coments

Validity of the findings

No Coments

Additional comments

The authors have made a good review about the English language and has attended to all raised questions satisfactorily. The work and findings are interesting and valuable for future genetic and evolutionary studies.

---

## Round 0.3 · accepted · Accept

All the reviewers' criticisms have been properly addressed.